# The Role of Exercise on Glial Cell Activity in Neuropathic Pain Management

**DOI:** 10.3390/cells14070487

**Published:** 2025-03-24

**Authors:** Willians Fernando Vieira, Caroline C. Real, Daniel Oliveira Martins, Marucia Chacur

**Affiliations:** 1Department of Anatomy, Institute of Biomedical Sciences (ICB), University of São Paulo (USP), 2415 Prof. Lineu Prestes Avenue, São Paulo 05508-000, SP, Brazil; willians.vieira@odonto.slmandic.edu.br; 2Department of Nuclear Medicine and PET Center, Aarhus University Hospital, 8200 Aarhus, Denmark; caroline.c.real@gmail.com; 3Laboratory of Neuroscience, Hospital Sírio-Libanês, São Paulo 3394-0200, SP, Brazil; danomartins82@gmail.com

**Keywords:** physical activity, physical exercise, exercise interventions, inflammation, pain, astrocytes, microglia, satellite glial cell, Schwann cells, enteric glial cells

## Abstract

Chronic pain is a widespread global health problem with profound socioeconomic implications, affecting millions of people of all ages. Glial cells (GCs) in pain pathways play essential roles in the processing of pain signals. Dysregulation of GC activity contributes to chronic pain states, making them targets for therapeutic interventions. Non-pharmacological approaches, such as exercise, are strongly recommended for effective pain management. This review examines the link between exercise, regular physical activity (PA), and glial cell-mediated pain processing, highlighting its potential as a strategy for managing chronic pain. Exercise not only improves overall health and quality of life but also influences the function of GCs. Recent research highlights the ability of exercise to mitigate neuroinflammatory responses and modulate the activity of GCs by reducing the activation of microglia and astrocytes, as well as modulating the expression biomarkers, thereby attenuating pain hypersensitivity. Here, we summarize new insights into the role of exercise as a non-pharmacological intervention for the relief of chronic pain.

## 1. Introduction

Chronic pain is a significant global health issue with a high prevalence worldwide and associations with conditions such as diabetes, cancer, fibromyalgia, and neurodegenerative diseases [1]. Neuropathic pain is defined as “pain caused by a lesion or disease of the somatosensory nervous system” [2]. In the United States of America (USA), an estimated 116 million adults experience chronic neuropathic pain, significantly impairing their quality of life and generating an annual societal cost of approximately $636 billion [3]. This condition arises from damage or dysfunction within the somatosensory system, leading to abnormal sensory signal transmission to the spinal cord and brain. Common conditions linked to neuropathic pain include diabetic neuropathy, peripheral nerve injuries, painful radiculopathy, trigeminal neuralgia, postherpetic neuralgia, HIV-related neuropathy, limb amputation, stroke, and cancer [1,4]. The pathways involved in pain perception constitute a complex sensory network that is activated by noxious stimuli, with neurotransmitters such as glutamate and substance P playing essential roles in signal transmission [5]. Within this network, several molecular and cellular pathways have been identified as key contributors to the pain process. Among these, the activation of glial cells (GCs) has emerged as a significant mechanism implicated in pain modulation and chronic pain development.

GCs are categorized into different types: astrocytes, oligodendrocytes, microglia, satellite glial cells (SGCs), Schwann cells (SCs), and enteric glial cells (EGCs). The activation of those cells is essential to damage tissue repair, restore the dysfunctional environment in the nervous system, or support and regulate neuronal function. In the context of pain processing, glial GCs have been increasingly recognized as key contributors [6]. Studies have shown that GCs significantly influence pain mechanisms by releasing neurotransmitters and inflammatory molecules that modulate pain pathways, impacting the experience and persistence of pain [7,8,9]. These cells interact bidirectionally with neurons, influencing pain perception and maintenance [10,11,12,13,14,15,16,17]. Chronic activation of GCs can lead to dysregulation of neuronal activity and alterations in synaptic plasticity, contributing to the development of chronic pain conditions such as neuropathic and inflammatory pain [18,19]. This central role in pain modulation makes GCs promising targets for therapeutic interventions in chronic pain management. Despite extensive efforts to control the pain process, in some cases, pain management remains a challenge [20,21,22]. A schematic illustration of the GCs and their role in neuropathic pain processing is shown in Figure 1.

There are several options to promote successful chronic pain treatment, including pharmacological and non-pharmacological approaches, for example, vagus nerve stimulation [23], physical exercise [24], and physical activity (PA) [25]. Here, it is important to highlight that PA is distinguished from physical exercise; PA encompasses all bodily movement, including daily routines, whether structured or unstructured, purposeful, or incidental. On the other hand, exercise is a specific subset of PA seeing that it refers to planned, structured, and repetitive movements to improve specific physical health outcomes (e.g., strength, cardiovascular endurance), whereas PA can be performed as part of daily life without specific health goals in mind. The PA can include daily activities like walking, gardening, cleaning, or even standing up from a chair, while physical exercise includes running, swimming, weightlifting, cycling, or participating in a fitness class frequently. To simplify, in this study, we will summarize both as exercise (EX).

According to the World Health Organization (WHO), adults should aim for 150 to 300 min of moderate-intensity aerobic EX per week (e.g., brisk walking, swimming, cycling), which is roughly 30 min a day, five days a week. Alternatively, they can engage in 75 to 150 min of vigorous-intensity aerobic EX per week (e.g., running, high-intensity interval training). A combination of moderate and vigorous-intensity exercise is also recommended to meet these guidelines. It should be noted that high-performance, intense exercise can potentially be harmful, primarily due to the formation of reactive oxygen species (ROS). These ROS can lead to oxidative stress and damage if not properly managed [26].

In this review, we will focus on chronic neuropathic pain, animal models, and clinical trials. Additionally, we will summarize important previous evidence on how different protocols of EX can modulate CG activation and improve chronic pain treatment. Exercise-induced hypoalgesia (ExIH) will also be discussed, as the mechanisms underlying its benefits are not yet fully understood. In addition, this review will highlight the relationship between non-pharmacological therapies, such as aerobic and strength EX, and how they may reduce neuropathic pain. Overall, the present review summarizes key evidence showing that EX influences neuroinflammation and GC activity, offering potential therapeutic benefits.

## 2. Exercise and Nervous System

Research on the health benefits of EX has gained significant momentum in the field of biomedicine over the past decade. It has been proposed that EX could be prescribed as a therapeutic intervention for various chronic diseases ((Pedersen and Saltin 2015a) [27]) and as a neuroprotective tool for neurological conditions. Among the health benefits of EX are improvements in metabolic and cardiovascular functions, reductions in adiposity, and enhancements in muscle tone and strength [28]. Additionally, EX improves cognitive and mental abilities, memory, mood, and symptoms of neurological conditions such as Alzheimer’s disease (AD), Parkinson’s disease (PD), and major depressive disorder (MDD) [28,29,30]. The mechanisms underlying these effects are not fully understood, however, there is evidence that EX activates anti-inflammatory pathways [27,28,31]. EX may also induce the expression of neurotrophic factors [32,33,34,35,36], increase blood flow and angiogenesis [37,38,39], and stimulate neurogenesis [40,41]. Enhanced brain function due to EX may, therefore, help reduce the expression of inflammatory markers, supporting the notion that inflammation plays a role in many neurological disorders [42]. On the other hand, the prevalence of physical inactivity presents a significant public health concern, as it contributes to increased all-cause mortality and fosters various health conditions, including obesity, cardiovascular disease, diabetes, cancer, and dementia [43,44,45,46,47]. Given these concerns, EX is an essential complementary strategy to pharmacotherapy in patients with degenerative diseases, helping to preserve cognitive function and independent living skills. This is particularly relevant in low-income countries, where medications to improve these symptoms are often expensive for most patients.

To better understand how exercise contributes to pain relief, numerous studies are being conducted with both animal models and human volunteers. Below, we describe some of these findings to elucidate the underlying mechanisms.

## 3. Exercise and Pain Management

In the last decades, there has been a growing body of evidence suggesting a positive correlation between a physically active lifestyle and health status. The literature consistently highlights the pivotal role EX plays in enhancing well-being, mitigating the risk of prevalent age-related conditions, such as cardiovascular diseases, stroke, and diabetes, while also offering relief from chronic pain. Non-pharmacological interventions present appealing alternatives to conventional analgesic treatments. Among these, EX, such as treadmill running or swimming, stands out. The interaction between GCs and EX represents a dynamic and multifaceted relationship that influences various aspects of the Central Nervous System (CNS) function, including pain modulation. Studies have demonstrated the supportive effects of astrocytes and microglia on neural cell proliferation and survival. Indeed, EX results in increasing activation and crosstalk between microglia and astrocytes. In general, EX modulates neuroinflammation by dampening the activation of microglia and reducing the production of pro-inflammatory cytokines and chemokines. By attenuating neuroinflammation, EX helps to alleviate neuropathic pain and promote CNS health [48,49,50].

While exercise demonstrates broad analgesic effects, certain neuropathic pain models respond differently to specific exercise interventions. In models of peripheral nerve injury (e.g., sciatic nerve constriction), treadmill running and swimming show significant reductions in pain behaviors, whereas, in diabetic neuropathy models, exercise outcomes are more variable, potentially due to metabolic complications influencing neuroinflammatory pathways. Similarly, in spinal cord injury models, the type and intensity of exercise play a crucial role, with excessive load sometimes exacerbating pain. These differences highlight the need for condition-specific exercise protocols rather than a one-size-fits-all approach.

### 3.1. Types and Intensity of Exercise: Effectiveness in Pain Management

Based on animal models, increasing the intensity of treadmill running (16 m/min compared with 10 m/min) resulted in greater analgesia, whereas increased training frequency (5 vs. 3 days/week) did not significantly affect analgesic effects [51]. Interestingly, combining treadmill training 2 weeks before and 2 weeks after injury yielded the most pronounced analgesic effects [52]. However, exercising to exhaustion counteracted the long-term analgesic benefits of EX, leading to more severe hyperalgesia than observed in sedentary, nerve-injured animals [48]. In EX models where animals are trained before the induction of neuropathic pain, results have been mixed. Some studies using voluntary wheel running for 6 weeks or treadmill training for 2 weeks have shown a reduction in the development of mechanical hyperalgesia [52,53]. In contrast, others found no effect when treadmill training was conducted 2 or 3 weeks before injury [54,55]. Interestingly, when animals are given access to running wheels after the induction of neuropathic pain, some studies demonstrate analgesic effects, while others do not show any ExIH [54,56]. Overall, most research indicates that various exercise paradigms have been effective in both preventing and alleviating neuropathic pain. Below, we summarize different types and intensity of exercise and their effectiveness in pain management (Table 1).

### 3.2. Exploration of the Mechanisms Underlying Exercise-Induced Pain Relief

Researchers have explored ExIH using a range of approaches involving animal models. To study ExIH, neuropathic pain models commonly employ methods that induce hyperalgesia, such as peripheral nerve injury, spinal cord damage, or diabetic conditions [57,58]. These models lead to both mechanical and thermal paw hyperalgesia, which can persist for weeks to months. EX protocols, whether initiated before or after the induction of neuropathic pain, have been shown to reduce hyperalgesia compared with sedentary animals [44].

Among these, aerobic EX protocols, such as treadmill running, swimming, or running wheel activity, are commonly studied [44]. According to Rice et al. (2019) [59], the analgesic effects of EX have been explored in animals by assessing alterations in their response to painful stimuli immediately after a single bout of EX, mirroring studies in human subjects examining ExIH following a single exercise session. Typically, this paradigm yields a brief analgesic effect lasting less than 30 min. The initial research showing ExIH in animals involved short sessions of a forced cold-water swim [60,61]. Those studies revealed that just 3 min of swimming decreased pain-related reactions to tail shocks and thermal tail-flick tests. However, distinguishing the ExIH from that induced by stress from the cold water was challenging in these early studies. As a result, contemporary research has shifted focus from cold-water swim tests to less stressful models [44,62]. Currently, EX models for animals include forced treadmill running, forced swimming, forced resistance training, and voluntary running wheels placed inside cages [44]. Forced treadmill running and swimming are popular and widely used exercise models since they allow precise control of the protocol, especially for intensity and duration. However, these models can increase stress biomarkers, which is important because stress can trigger pain relief by activating opioid and serotonergic systems. This adds complexity to interpreting study results, as stress effects may confound the direct influence of ExIH [63,64,65]. To mitigate this issue, some researchers use voluntary running wheels, which enable animals to exercise without elevating stress levels. However, this model also has limitations, as each animal exercises differently, increasing data variability.

Forced exercise, such as treadmill running, has been identified as a potential stressor that may independently alter glial activation and pain thresholds. Studies measuring stress biomarkers, such as corticosterone levels, have found that forced exercise induces a greater stress response compared to voluntary exercise [66,67]. This may partially explain why some studies report increased hyperalgesia following exhaustive treadmill training. Controlling for stress markers is essential to distinguish whether the observed analgesic effects result from exercise itself or from stress-induced activation of endogenous pain modulation pathways. Microglia exhibit a rapid response to stressful events, including acute [68,69] and repeated stress [70,71,72,73]. One study demonstrated that stress-induced microglial activation occurs within 30 min of exposure to a stressor [68] and that this activation was completely blocked by propranolol, a β1- and β2-adrenergic receptor antagonist [74]. These findings suggest that microglia may receive noradrenergic signals in stressed brains, which is particularly relevant given that central β-adrenergic receptors (β-ARs) also play a role in the beneficial effects of exercise [75,76,77,78]. This highlights the reciprocal relationship between stress and exercise. However, the molecular mechanisms underlying stress-related microglial activation remain to be fully elucidated.

Despite these challenges, most EX models have shown analgesic effects in pain-free animals and have proven effective in preventing or reversing hyperalgesia in various animal models, including those for neuropathic, inflammatory, and non-inflammatory muscle pain [44].

For instance, repetitive swimming protocols involving 50, 60, or 90 min of swimming, five days a week, reverse both mechanical and thermal hyperalgesia in animals with peripheral nerve injuries after the onset of EX [79,80]. Similarly, treadmill exercise protocols have demonstrated reductions in both mechanical and thermal hyperalgesia in animal models of neuropathic pain [52,81,82]. These treadmill protocols, which include repeated bouts of 10, 20, 30, or 60 min, ranging from 3 to 7 days a week, have effectively reduced hyperalgesia when initiated before or after the induction of nerve injury. Additionally, starting EX either 1 week or 3 weeks after nerve injury led to analgesia 2 weeks after the beginning of training [51]. This suggests that EX is effective in reducing pain regardless of when it is initiated (after or before the injury).

Although EX is a promising treatment approach for chronic pain conditions, many questions remain about its implementation. Specifically, uncertainties persist regarding the most effective exercise modalities, duration, intensity, and volume for individuals with chronic pain. Clarifying these mechanisms underlying ExIH holds the potential to refine EX protocols for chronic pain management and facilitate the identification of novel pharmacological targets for pain relief [44].

It has been suggested that regular EX alters the state of central pain inhibitory pathways and the immune system, leading to a protective effect against peripheral insults. This protective state, which occurs with regular EX, is absent in physically inactive individuals, increasing their risk of developing chronic, long-lasting pain [83]. It has also been suggested that EX modulates the immune system at the site of insult, both systemically and within the CNS [84,85]. In physically inactive individuals, there are elevated levels of inflammatory cytokines and reduced levels of anti-inflammatory cytokines [86]. Regular EX shifts the balance toward increased levels of anti-inflammatory cytokines and decreased levels of inflammatory cytokines. Inflammatory cytokines activate receptors on nociceptors, leading to pain, whereas anti-inflammatory cytokines reduce nociceptor activity, thereby helping to prevent pain [87,88].

In addition to changes in the level of inflammatory markers, microglia can present two different phenotypes with opposite functions, which can shift dynamically depending on the stimulus: (1) harmful M1 phenotype, releasing proinflammatory markers, and (2) beneficial M2 phenotype, releasing anti-inflammatory markers. It will be further described below, in Section 4.1.

Several studies have shown that EX can protect the nervous system from a toxic environment by enhancing neurogenesis and reducing inflammation [89,90,91], as happens in Alzheimer’s (AD) and Parkinson’s disease (PD) pathology [89,90,91,92]. However, its impact on spinal microglia in neuropathic pain is less explored [92]. Preoperative exercise has been shown to decrease postoperative pain and improve recovery in the first six months, but its benefits seem to wane afterward [93,94]. Regular EX can modulate immune responses and may alleviate chronic pain by providing anti-inflammatory effects, promoting neurogenesis, and offering temporary pain relief. Notably, evidence-based practice guidelines endorse exercise as an intervention for various chronic pain conditions, such as low back pain, osteoarthritis, and fibromyalgia, supported by moderate to strong evidence [52,81]. Although acute exercise bouts may initially exacerbate pain in chronic pain sufferers and animal models, extensive research demonstrates that regular EX plays a dual role in both preventing and alleviating chronic pain, substantiating their therapeutic potential.

Other factors involved in pain relief include neurotrophins [58,95], opioids [96,97], and endocannabinoids [98]. EX activates the endogenous opioid system, leading to the release of endorphins and enkephalins, which act as natural painkillers. These opioid peptides interact with opioid receptors on glial cells, including microglia and astrocytes, modulating their activity, and suppressing pro-inflammatory responses [51,99,100]. By engaging the endogenous opioid system, exercise helps to alleviate pain and reduce neuroinflammation. Overall, the interaction between GCs and EX represents a vital mechanism through which exercise exerts its beneficial effects on pain modulation and CNS function. Although the role of the endocannabinoid system in exercise-induced reductions in neuropathic pain has not yet been thoroughly investigated, evidence suggests that endocannabinoids play a key role in modulating nociceptive pain responses [24]. Prior studies from our group have shown that endocannabinoid signaling can mitigate pain responses in animal models of Parkinson’s disease [101,102,103,104], supporting the potential of endocannabinoid pathways as therapeutic targets. This evidence underscores the need for further research to explore whether endocannabinoids also contribute to the pain-relieving effects of exercise in neuropathic pain contexts.

Inflammatory mediators released from activated glia contribute to neuroinflammation, a hallmark of many chronic pain conditions. Neuroinflammation involves the infiltration of immune cells into the CNS, including macrophages and lymphocytes, further amplifying the inflammatory response. Chronic neuroinflammation not only sensitizes nociceptive pathways but also promotes neuronal hyperexcitability and synaptic plasticity, leading to persistent pain states [105,106,107,108]. Furthermore, GCs can communicate bidirectionally with neurons through several signaling pathways, including purinergic, glutamatergic, and chemokine signaling [7,109]. Additionally, GCs can respond to neurotransmitters released by neurons, thereby further shaping neuronal excitability and synaptic transmission [109]. Disruption of this intricate communication contributes to aberrant pain signaling and the development of chronic pain conditions. Targeting glial activation, neuroinflammation, and glial-neuronal interactions holds promise for the treatment of chronic pain [7,109,110,111].

It has been reported that inhibition of brain glial cell activation significantly reduces neuroinflammation and brain edema in mice intoxicated with 1,2-dichloroethane, highlighting the critical role of astrocyte–microglia crosstalk in these pathological processes [112,113]. On the other hand, substances that inhibit astrocyte function are effective in both the early and late stages of neuropathic pain, suggesting that microglia and astrocytes play different roles in the onset and persistence of chronic pain [114,115]. Studies indicate that microglia activation can trigger astrocyte activation and vice-versa. This glial activation leads to the release of gliotransmitters, such as glutamate and ATP, which further sensitize neurons and perpetuate neuropathic pain [116,117,118]. Furthermore, in the process of inflammatory pain and neuropathic pain, microglia and astrocytes release inflammatory mediators, including prostaglandins and interleukins, which sensitize neurons and contribute to the development and maintenance of these pain states [119,120,121]. Studies have demonstrated that direct stimulation of astrocytes induces a reactive response in microglia, leading to increased sensitivity to pain. Furthermore, suppressing astrocyte activity by removing a specific receptor (CXCR5) reduces microglial activation in neuropathic pain conditions [122]. In cases of neurological pain, there is an increase in CB2 mRNA in the activated microglia of nerve roots [123]. Particularly, CB2 receptor activity is increased in the dorsal horn in a variety of neurological pain scenarios, involving regional damage to neurons, chemotherapy-induced neurological pain, and persistent post-ischemia pain. This increased expression is associated with microglia that have been activated [124]. In chronic constriction injury (CCI) of the sciatic nerve, a model of neuropathic pain causes depression-like behavior in animals, and the cannabinoid CB2-selective agonist GW405833 improves immobility and mechanical hypersensitivity, reinforcing the role of the CB2 pathway in nociceptive control in animal models [125]. This suggests that the beneficial effects of exercise on neurological symptoms may be mediated, at least in part, through its influence on microglial activity [92]. The beneficial effects described earlier involve various cell types from the immune system of the nervous systems and are illustrated in the image below (Figure 2). To provide further clarity, we will now detail the role of each cell type in pain relief.

These studies suggest that regular exercise can not only prevent or alleviate pain but may also modulate the immune system, shift the balance of inflammatory cytokines, and influence neurogenesis. Additionally, exercise can impact the function of glial cells, such as microglia and astrocytes, which are central to neuroinflammation and chronic pain. One important aspect that stands out in the research is the dual role of exercise, both providing temporary pain relief in the short term and offering long-term benefits, such as neuroprotection and a reduction in chronic pain states. The modulation of the immune system through exercise is also key, as it enhances anti-inflammatory cytokine release while reducing pro-inflammatory markers. The exercise’s ability to engage the endogenous opioid system and reduce neuroinflammation could explain its therapeutic potential in managing chronic pain.

The involvement of glial cells, particularly microglia, in the neuroinflammatory response associated with chronic pain highlights the complexity of pain modulation. Research suggests that glial–neuronal interactions and the activation of glial cells, such as astrocytes and microglia, play significant roles in the persistence of chronic pain. Furthermore, the potential role of endocannabinoids in modulating pain through these glial cells suggests a promising direction for future research in understanding and treating pain, especially in contexts of neurological conditions.

## 4. Exercise and Glial Cells

EX induces significant effects on GCs, which play a vital role in the CNS’s response to physical activity. These cells are involved in regulating neuroinflammation, maintaining neuronal homeostasis, and supporting synaptic plasticity. During and after EX, GCs, particularly astrocytes and microglia, modulate the inflammatory response and contribute to tissue repair and recovery. The adaptation of GCs to EX is crucial for enhancing neural function, mitigating EX-induced damage, and promoting overall neuroplasticity and pain modulation. In the following sections, we will highlight key aspects of the effects of EX on each type of GC individually.

### 4.1. Microglia

EX has been increasingly recognized for its potential to modulate GC function, thereby offering relief from pain. However, the role of CNS glia on EX and pain is not entirely clear (for discussion, see [95,100]. Activation of these cells is characterized by marked changes in their number, morphology, gene expression, and function, leading to the release of trophic factors, cytokines, and chemokines.

It is well known that during injury or inflammation, microglia undergo activation, releasing pro-inflammatory cytokines and chemokines, which contribute to the amplification of pain signals. EX using a treadmill at 20 m/min for 30 min, 5 days per week for 5 weeks post-chronic constriction injury in rats has been demonstrated to decrease microglial activation, thereby reducing the release of these inflammatory mediators [49]. The activation of these GCs sensitizes dorsal horn neurons through various mechanisms, including the release of pro-nociceptive molecules, such as IL-1β, IL-6, TNF-α, NO, prostaglandin, endocannabinoids, chemokine, and BDNF. For instance, Chen et al. 2012 [57] observed significant pain reduction in rats subjected to CCI after engaging in treadmill running (5 days a week for 6 weeks) or swimming (90 min per session), attributing this effect to decreased levels of TNF-α and IL-1β. Similar findings were observed by Bobinski et al. (2011) [52], who found that treadmill running (30 min at a speed of 10 m/min, no inclination, 5 days per week) alleviated pain in a mouse model of crushed sciatic nerve, both pre- and post-injury.

In the same line of reasoning that EX intensity should be an important factor for improving outcomes, studies conducted by Cobianchi et al., (2010) [48] have shown that short-lasting EX (1 h/d for 5 days after CCI) induces pain relief and stimulates nerve regeneration after peripheral nerve injury. The reduced expression of microglia in short-lasting trained mice, followed by at least two-fold reduction of astrocytes, indicates that short-lasting treadmill running was sufficient not only to reduce microglia but also to reduce the astrocytes expression when compared to long-lasting running (1 h/d for more than 5 days after CCI) [48].

Upon activation, microglia exhibit two distinct states, known as M1 and M2 phenotypes [126]. The M1-like phenotype is triggered by inflammatory agents like TNF-α and IL-1β, prompting microglia to release pro-inflammatory molecules such as IL-6, IL-23, and chemokines. This initiates neuroinflammation, contributing to the persistence of chronic pain. Conversely, the M2-like phenotype is induced by cytokines such as IL-4 or IL-13. M2-like microglia exhibit increased phagocytic activity and produce growth factors like insulin and insulin-like growth factor-1 (IGF-1), as well as anti-inflammatory cytokines such as IL-10. Communication between astrocytes and microglia is mediated through the secretion of various molecules, including growth factors, cytokines, chemokines, ATP, NO, ROS, and metabolic intermediates such as amino acids. These molecules play crucial roles in cellular metabolic functions and intercellular signaling [112,127].

According to Sluka et al. (2018) [83], in the periphery, EX levels influence the phenotype of macrophages in muscle tissue. Macrophages can release either inflammatory or anti-inflammatory cytokines depending on their type. Activated (M1) macrophages secrete inflammatory cytokines, while regulatory (M2) macrophages release anti-inflammatory cytokines. Consistent with this, physically active animals, whether uninjured or with nerve injury, show a higher proportion of M2 macrophages compared to sedentary animals. Following nerve injuries, sedentary animals have a higher proportion of M1 macrophages and a lower proportion of M2 macrophages at the injury site. In contrast, physically active animals show an increase in M2 macrophages and a decrease in M1 macrophages. The analgesic effects of EX are hindered by blocking either IL-10 (in cases of muscle insult) or IL-4 (in cases of nerve injury). This suggests that alterations in macrophage phenotype at the site of insult play a key role in ExIH.

EX has been found to enhance astrocytic function, including the clearance of neurotransmitters like glutamate, which can contribute to pain hypersensitivity when present in excess. By promoting astrocytic support, EX contributes to the maintenance of synaptic homeostasis and pain modulation [49]. EX stimulates the production and release of neurotrophic factors such as BDNF. These factors not only support neuronal survival and plasticity but also regulate GC function. BDNF, in particular, has anti-inflammatory properties and can modulate microglial and astrocytic activity, contributing to pain relief [128].

In addition to the role of EX in preventing microglia from becoming over-activated, it is also important in maintaining microglial phenotype homeostasis. Microglia can present two different phenotypes with opposite functions that can shift dynamically, depending on the stimulus: (1) harmful M1 phenotype, releasing proinflammatory markers, and (2) beneficial M2 phenotype, releasing anti-inflammatory markers. In this sense, there are studies showing that EX can alleviate neurological symptoms through mechanisms to stimulate the shift from M1 to M2 phenotype [92,129,130]. EX-induced alterations in astrocyte morphology, including increased branching and greater synaptic coverage, may play a key role in enhancing synaptic plasticity and modulating pain perception [130]. Additionally, GCs, particularly astrocytes, are crucial in maintaining extracellular glutamate levels. Studies have shown that EX enhances glutamate clearance from the synaptic cleft by astrocytes, thereby reducing excitotoxicity and dampening pain signaling [131,132]. In addition, Almeida et al. (2015) discovered that extending the duration of swimming activity reversed the overactivity of astrocytes and microglia in the dorsal horn of the spinal cord following nerve injury, thereby reducing pain behavior in mice [58].

### 4.2. Astrocytes

Astrocytes play a crucial role in pain processing by regulating synaptic transmission and neuronal excitability. When astrocyte activity becomes dysregulated, it can contribute to the development and persistence of chronic pain conditions. EX has been found to promote a more balanced astrocytic phenotype, reducing the release of pain-promoting factors while enhancing the secretion of neurotrophic factors that support neuronal health.

It was reported that endurance exercise (treadmill for 30 min three times a week for 4 weeks, no inclination) reduced glial fibrillary acidic protein (GFAP) expression in brain astrocytes in a rat model of cerebral palsy [133]. This indicates a reduction in astrocytic activation, but changes in pain were not reported. In a separate study, the authors assessed the impact of three forms of overtraining—downhill running, uphill running, and overtraining without inclination—over an 8-week period, comprising five days of training followed by two days of recovery. They found that spinal cord GFAP levels exhibited fluctuations, either increasing or decreasing, depending on the type of EX undertaken [134].

The high-frequency EX (treadmill at a speed of 20 m/min, 5 days) and low-frequency EX program (3 days) for a total of 5 weeks indicated that aerobic EX promoted ExIH through a reduction in GFAP (an astrocyte marker) and ionized calcium-binding adapter molecule 1 (Iba1, a microglia marker) in the spinal cord [135]. The authors suggested a relationship between the frequency of EX and glial activation and BDNF. Four weeks of treadmill induces the production and release of various neurotrophic factors, such as BDNF, which exert potent anti-inflammatory and analgesic effects, decreasing pain [92]. The alleviated pain observed by the authors suggests that EX may regulate spinal nerve ligation-induced (SNI) astrocyte reactivity and normalize immune changes through suppressed astroglial C3 expression, resulting in reduced IL-1β and TNF-α in the spinal cord of mice.

Using an SNI model, Stagg et al. (2011) [51] reported that EX training for 3 days per week did not improve the tactile paw withdrawal threshold to the same level as EX training for 5 days per week. This demonstrated that the intensity of EX, rather than its frequency, resulted in a more complete recovery in a model of neuropathic pain. Animal studies have shown a decrease in astrocytes after moderate treadmill exercise (20 min, 5 days a week for 4 weeks) [136], suggesting that changes in astrocyte markers induced by EX reinforce its relevance as a strategy for neuroprotection.

These factors not only support neuronal survival and plasticity but also help regulate GC function. By enhancing the expression of neurotrophic factors, EX creates an environment conducive to pain resolution and neuronal repair. The frequency of EX should be taken into consideration when creating a physical therapy program for the rehabilitation of patients with neuropathic pain.

### 4.3. Oligodendrocytes

Oligodendrocytes play a crucial role in maintaining the structural integrity and function of neurons by producing myelin, a fatty substance that wraps around nerve fibers to insulate and facilitate efficient signal transmission. While research on the direct interaction between EX and oligodendrocytes is still emerging, there is evidence to suggest that EX may influence oligodendrocyte function and myelination in the CNS [137]. Research suggests that EX may stimulate oligodendrogenesis, the process by which new oligodendrocytes are generated from precursor cells. Regular EX has been linked to elevated levels of growth factors and signaling molecules, such as IGF-1 and BDNF, which promote oligodendrocyte development and support myelination. Myelination is essential for the rapid and efficient transmission of electrical impulses along axons. EX has been shown to enhance myelination in various regions of the brain, including the hippocampus and motor cortex, which are involved in learning, memory, and motor function, respectively. This suggests that EX-induced changes in oligodendrocyte function may contribute to the structural and functional adaptations observed in the CNS.

Glial modulation plays a central role in exercise-induced analgesia, connecting mechanisms such as endogenous opioids, endocannabinoids, cytokines, and BDNF within a coordinated response that involves both systemic and neural adaptations (Figure 3). When exercise is performed, it triggers the release of molecular signals that affect both neurons and glial cells, promoting an anti-inflammatory and analgesic response. Glial cells, particularly microglia and astrocytes, respond to exercise by modulating neurotransmitters and cytokines, which, in turn, influence neuronal activity. The release of endogenous opioids and endocannabinoids during exercise has a direct effect on specific receptors in the central nervous system, resulting in pain reduction. Anti-inflammatory cytokines and BDNF are also involved in this process, promoting neuroprotection and neuronal plasticity. Thus, glial modulation not only facilitates the activation of these systems but also orchestrates the integration of local and systemic responses, ensuring long-term neural adaptation and reduced pain perception. This understanding of how glial modulation fits into the broader landscape of exercise-induced adaptations suggests that manipulating this pathway could optimize the therapeutic effects of exercise on neuropathic pain and other related disorders.

## 5. Clinical Implications and Future Directions

Although there is growing evidence supporting the safety and efficacy of EX, its impact on GC modulation remains incompletely understood, and no specific guidelines exist for its optimal use in neuropathic pain management. Implementing EX as a non-pharmacological therapeutic strategy carries significant clinical implications for both human and animal models. In both clinical and experimental settings, EX has been shown to modulate pain perception and enhance functional outcomes. This is particularly relevant when considering chronic pain conditions, which often present a complex interplay of physiological, psychological, and social factors [83,107,138,139]. In clinical practice, EX interventions are increasingly recognized as integral components of multidisciplinary pain management strategies. EX programs personalized to individual needs and capabilities can help improve physical function, reduce pain intensity, and enhance the overall quality of life for individuals living with chronic pain. However, engaging in EX protocols is especially difficult for patients with neuropathic pain, as most are insufficiently active or sedentary. In experimental models, EX has been extensively studied to elucidate its mechanisms of action in pain modulation, and ongoing research is needed to deepen our understanding of the molecular bases underlying the impact of EX on patients’ well-being.

One area of growing interest is the involvement of GCs, both in the CNS and the PNS, in mediating the effects of EX on pain threshold. GCs, including microglia and astrocytes in the CNS, and SGCs in the PNS, play crucial roles in neuroinflammation and pain processing [92,140]. Understanding the role of GCs in mediating the effects of EX on pain perception could pave the way for personalized EX prescriptions tailored to individuals with chronic pain. Combining EX interventions with pharmacological treatments that target glial function may yield synergistic effects, enhancing pain relief while reducing the need for higher medication doses and minimizing associated side effects.

Further research is needed to elucidate the specific molecular mechanisms through which EX modulates glial function. Investigating signaling pathways and gene expression patterns involved in EX-induced changes in glial activity could provide insights into new therapeutic targets for pain management. Investigating the long-term effects of exercise on glial function and pain outcomes is crucial to understanding EX-based interventions in the treatment of chronic pain. Longitudinal studies evaluating the durability of glial modulation by EX and its impact on pain recurrence rates could inform the development of maintenance strategies for sustained pain relief.

### Current Clinical Evidence and Research Gaps

While numerous preclinical studies provide strong evidence for the analgesic effects of exercise in neuropathic pain, the translation of these findings into clinical guidelines remains incomplete. Some preliminary clinical trials suggest that structured exercise programs may improve pain management in individuals with diabetic neuropathy [141], post-surgical nerve pain [142], and chemotherapy-induced neuropathy [143]. Specifically, moderate-intensity aerobic exercise (e.g., walking, cycling) has been associated with reduced pain intensity, improved functional mobility, and lower levels of pro-inflammatory cytokines. Additionally, emerging evidence suggests that resistance training may provide benefits by enhancing neuromuscular function and reducing pain-related disability. However, despite these promising results, current clinical practice guidelines lack standardized exercise parameters for neuropathic pain management. Studies vary widely in terms of exercise type, intensity, frequency, and duration, making it difficult to draw definitive conclusions on optimal protocols. Moreover, patient-specific factors—such as pain severity, comorbidities, and individual physical capacity—must be considered when designing exercise interventions. Given these gaps, further randomized controlled trials are needed to establish evidence-based exercise recommendations tailored to different neuropathic pain conditions.

## 6. Conclusions

Despite significant efforts by researchers to understand pain mechanisms and develop effective relief strategies for this debilitating condition, gaps remain. Future directions in pain management should focus on elucidating the specific molecular and cellular mechanisms through which EX influences glial function and pain processing. Additionally, integrating translational approaches that connect findings from preclinical animal models to clinical populations will be crucial for optimizing exercise-based interventions for pain management.

## Figures and Tables

**Figure 1 cells-14-00487-f001:**
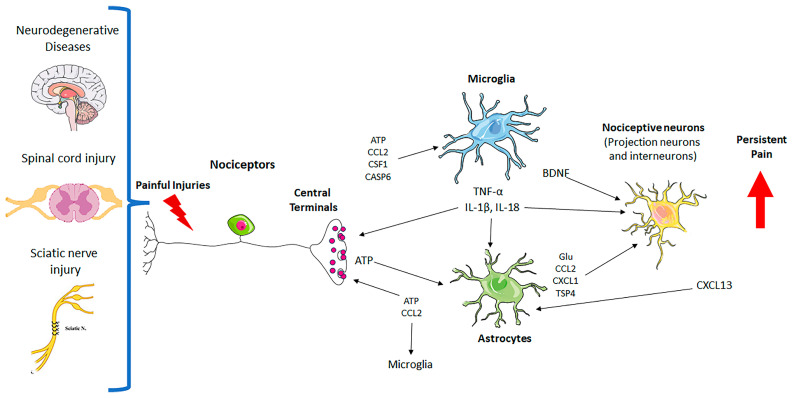
Neuron–glial interactions in persistent pain. Painful injuries, such as neurodegenerative diseases, spinal cord injury, and sciatic nerve injury, cause hyperactivity of nociceptors and secretion of glial modulators from their central terminals, leading to the activation of microglia and astrocytes in the spinal cord dorsal horn. Upon activation, microglia and astrocytes secrete neuromodulators to drive chronic pain by inducing synaptic and neuronal plasticity. CASP6: caspase-6; Glu: glutamate; TNF-α: tumor necrosis factor; IL-1β: interleukin-1β; ATP: adenosine 5′-triphosphate; TSP4: thrombospondin-4; BDNF: brain-derived growth factor; CSF: colony-stimulating factor-1; chemokines (CCL2, CXCL1, CXCL13).

**Figure 2 cells-14-00487-f002:**
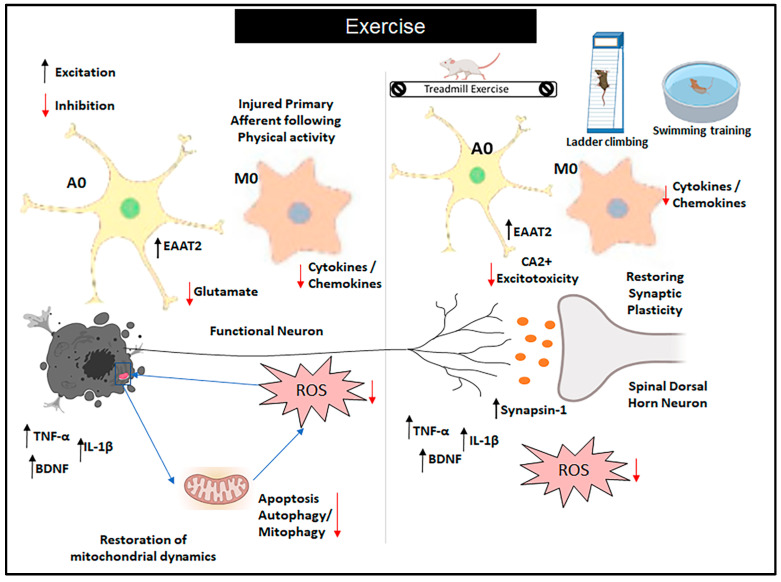
Role of exercise on glial cells in neuropathic pain. Exercise (EX) is recommended for the treatment of neuropathic pain, and its effectiveness has been linked to the inhibition of glial cell (GC) activation and modulation of inflammation in both patients and animal models of neuropathic pain. This is achieved by downregulating the expression of proinflammatory cytokines and upregulating the expression of anti-inflammatory cytokines. Additionally, evidence suggests that EX promotes the upregulation of trophic factors, which subsequently leads to reduced neuronal injury and decreased activation of GCs. Abbreviations: A0: resting astrocytes; M0: non-activated microglia; BDNF: brain-derived growth factor; EAAT2: excitatory amino acid transporter 2; Ca^2+^: calcium excitotoxicity; TNF-α: tumor necrosis factor; IL-1β: interleukin-1β; ROS: reactive oxygen species.

**Figure 3 cells-14-00487-f003:**
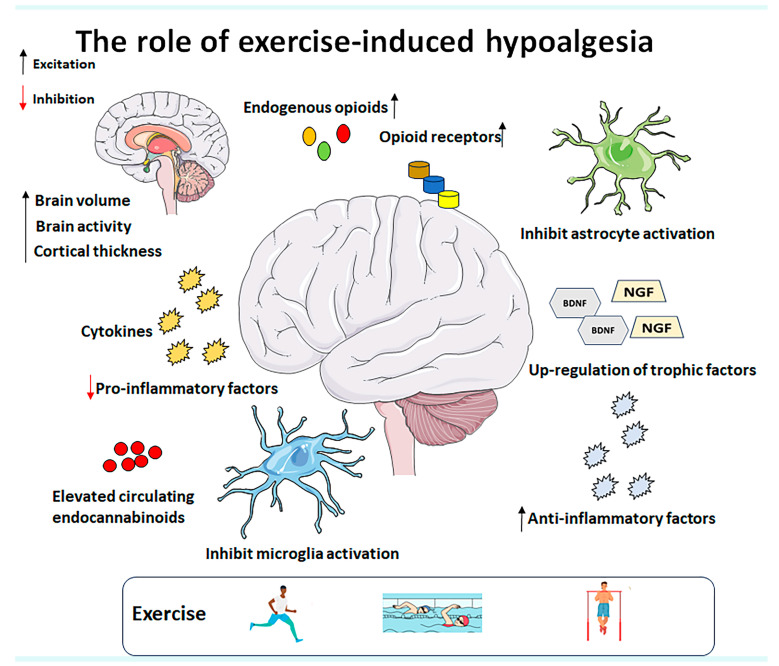
Potential mechanisms underlying exercise-induced hypoalgesia include the modulation of endogenous opioids, upregulation of trophic factors, increased production of anti-inflammatory cytokines, inhibition of pro-inflammatory factor expression, suppression of microglia and astrocyte activation, and elevated levels of circulating endocannabinoids. Additionally, exercise induces structural and functional changes in the brain. Therefore, further clinical studies with larger sample sizes and well-defined protocols are needed to explore the specific parameters of different exercise therapies for various pain syndromes. Abbreviations: BDNF (brain-derived neurotrophic factor) and NGF (nerve growth factor).

**Table 1 cells-14-00487-t001:** Summary of key exercise protocols and outcomes.

Exercise Protocol	Model	Intensity	Duration	Analgesic Effects	Conflicting Findings
Treadmil (10–16 m/min)	Nerve Injury (CCI)	Moderate	2–6 weeks	Reduces hyperalgesia, lowers pro-inflammatory cytokines	Higher speeds show greater effect, but exhaustive running worsens pain
Swimming (60–90 min)	Nerve injury, diabetic neuropathy	High	3–5 weeks	Reduces pain behaviors, enhances neurotrophic factors	Cold stress may confound effects
Voluntary wheel running	Neuropathic pain	Self-paced	4–6 weeks	Mixed results; some studies show benefit, others show no effect	High variabillity in running duration per animal

## Data Availability

No new data were created or analyzed in this study.

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
