# Peer review of "The Role of Exercise on Glial Cell Activity in Neuropathic Pain Management"

_cells, 2025, doi:10.3390/cells14070487_

Round 1

Reviewer 1 Report

Comments and Suggestions for Authors

The topic of Review is of wide interest and deserved to be published but not in the modern shape. It needs serious changes and improvement mainly in the way of presentation.

To begin with Title that has a lot of unnecessary words e.g. “interventions” , “ modulation”;  “Exploring” is more suitable for experimental study not review  - should be changed.

Some more examples

Abstract:  l.16 – “In this sense…” - recommended to be deleted.

  1. 18 – “its potential’ – whose potential?

Similar comments may be found all over the text and should be fixed before publication.

l.119 “Research into the potential health…” this sentence should be rewritten.

l.170 – not clear what injury was used.

l.306 – “Inhibition of microglia and astrocytes…”  what does this mean

Similar examples of unclear and controversial  expressions are all over the text and should be fixed.

Figure 1. Recommended to change profile of mouse brain for profile of human brain.

Comments on the Quality of English Language

English grammar is correct but the logic presentation is very poor

Author Response

1.The topic of Review is of wide interest and deserved to be published but not in the modern shape. It needs serious changes and improvement mainly in the way of presentation.

To begin with Title that has a lot of unnecessary words e.g. “interventions” , “ modulation”;  “Exploring” is more suitable for experimental study not review  - should be changed.

Answer: We thank the reviewer for their valuable feedback. We understand your concern about the title and appreciate your suggestions. We agree that simplifying the title would improve its clarity and precision. Based on your feedback, we propose the revised title: “The Role of Exercise on Glial Cell Activity in Neuropathic Pain Management”. We hope this addresses your concerns and are open to further suggestions.

2.Some more examples

Abstract:  l.16 – “In this sense…” - recommended to be deleted.

18 – “its potential’ – whose potential?

Answer: Regarding the abstract, we agree that the phrase 'In this sense...' can be removed to improve clarity and conciseness. As for line 18, we have clarified the subject of 'its potential'. We have made these adjustments in the revised manuscript.

3.Similar comments may be found all over the text and should be fixed before publication.

l.119 “Research into the potential health…” this sentence should be rewritten. l.170 – not clear what injury was used. l.306 – “Inhibition of microglia and astrocytes…”  what does this mean

Similar examples of unclear and controversial  expressions are all over the text and should be fixed.

Answer: As suggested by the reviewer, the mentioned sentences were fixed to improve clarity.

4.Figure 1. Recommended to change profile of mouse brain for profile of human brain.

Answer: Thank you for your suggestion. The change in Figure 1 was made accordingly.

5.Comments on the Quality of English Language

English grammar is correct but the logic presentation is very poor

Answer: We improve the English language

Reviewer 2 Report

Comments and Suggestions for Authors

The manuscript, "Exploring the role of exercise interventions on glial cells activity modulation in neuropathic pain management," reviews the current evidence on how different exercise (EX) regimens affect glial cell (GC) activity-primarily microglia and astrocytes-and how these changes might contribute to the prevention or relief of neuropathic pain. Overall, this work contributes to a growing body of literature that highlights the potential of exercise as a nonpharmacologic intervention for neuropathic pain. The review is timely, as chronic pain management increasingly embraces multimodal and lifestyle-based approaches.

The following are proposed key gaps to be addressed:

  1. While the review briefly mentions clinical implications, it does not provide concrete or detailed information on how preclinical findings might translate into specific exercise recommendations for humans with neuropathic pain. Include a brief subsection summarizing any existing clinical trials or practice guidelines that address exercise parameters for neuropathic pain, or at least acknowledge the gap if these data are lacking.
  2. The review covers a wide range of intensities and protocols (e.g., treadmill speeds, durations, pre- vs. post-injury training), but does not thoroughly compare or clearly state which protocols show the most consistent benefit or why some are inconclusive. Provide a brief, comparative summary table (or short paragraph) that identifies the major protocols used in key studies, the analgesic outcomes observed, and any conflicting results.
  3. Some studies show no effect of exercise when performed before injury, while others report significant benefits of pre-injury exercise. Similarly, exhaustive exercise sometimes worsens pain, but moderate exercise is protective. These contradictions are mentioned but not well integrated. The review should help interpret or at least acknowledge possible reasons (e.g., different load levels, intensity thresholds, methodological differences). Provide a clear, brief discussion of potential confounders - such as stress biomarkers, sex differences, timing of exercise initiation - to help reconcile or emphasize these inconsistencies.
  4. The review acknowledges that forced exercise (e.g., treadmill) can be stressful. However, it does not address how stress alone, independent of the exercise regimen, might affect glial activation or pain thresholds. Briefly highlight any studies that have controlled for or measured stress markers and how this might confound results.
  5. The text lists several analgesic mechanisms (endogenous opioids, endocannabinoids, cytokines, BDNF) but does not tie them together into a cohesive understanding of how glial modulation fits into the bigger picture of exercise-induced hypoalgesia. Provide a brief subsection or schematic that places glial modulation within the broader landscape of exercise-induced systemic and neural adaptations.
  6. The manuscript references multiple neuropathic pain models - nerve injury, diabetic neuropathy, spinal cord injury - without distinguishing whether specific exercise strategies appear to be more or less beneficial for specific etiologies. If available, highlighting any model-specific findings or consistent patterns would give readers insight into whether exercise is universally beneficial or more relevant to specific neuropathic conditions.
  7. Ensure that any aberritives used for the first time are explained, including Table 1 (CCI).

Addressing these points - particularly clarifying exercise variables, expanding on conflicting findings, and providing a more coherent mechanistic overview - would greatly improve the review.

Author Response

1-The manuscript, "Exploring the role of exercise interventions on glial cells activity modulation in neuropathic pain management," reviews the current evidence on how different exercise (EX) regimens affect glial cell (GC) activity-primarily microglia and astrocytes-and how these changes might contribute to the prevention or relief of neuropathic pain. Overall, this work contributes to a growing body of literature that highlights the potential of exercise as a nonpharmacologic intervention for neuropathic pain. The review is timely, as chronic pain management increasingly embraces multimodal and lifestyle-based approaches.

The following are proposed key gaps to be addressed:

While the review briefly mentions clinical implications, it does not provide concrete or detailed information on how preclinical findings might translate into specific exercise recommendations for humans with neuropathic pain. Include a brief subsection summarizing any existing clinical trials or practice guidelines that address exercise parameters for neuropathic pain, or at least acknowledge the gap if these data are lacking.

Answer: Thank you for your insightful comment. We agree that a more detailed discussion of the translation from preclinical findings to clinical practice would strengthen the review. To address this, we have included a brief subsection under 5.1. Current Clinical Evidence and Research Gaps summarizing existing clinical trials and guidelines related to exercise for neuropathic pain. Since clinical data is limited, we explicitly acknowledge this gap and emphasize the need for further research. Briefly, although numerous preclinical studies support the analgesic effects of exercise in neuropathic pain, clinical guidelines on specific exercise prescriptions remain limited. However, some studies have provided preliminary evidence for exercise as a therapeutic strategy in neuropathic pain conditions. Clinical trials investigating structured exercise programs in patients with diabetic neuropathy and post-surgical nerve pain have reported reductions in pain intensity and improvements in functional mobility. Notably, moderate aerobic exercise (e.g., walking, cycling) has been associated with decreased pro-inflammatory cytokine levels and enhanced endogenous pain modulation in chronic pain patients. Despite these promising findings, there is a need for more randomized controlled trials to establish standardized exercise parameters tailored to different neuropathic conditions.

2-The review covers a wide range of intensities and protocols (e.g., treadmill speeds, durations, pre- vs. post-injury training), but does not thoroughly compare or clearly state which protocols show the most consistent benefit or why some are inconclusive. Provide a brief, comparative summary table (or short paragraph) that identifies the major protocols used in key studies, the analgesic outcomes observed, and any conflicting results.

Answer: Thank you for your valuable suggestion. We acknowledge the importance of clearly comparing the exercise protocols and their outcomes. To address this, we have included a comparative summary table under  3.1.Types and Intensity of Exercise to systematically present the major exercise protocols investigated in preclinical and clinical studies. This table highlights key variables such as exercise modality, intensity, duration, and observed analgesic effects, while also addressing inconsistencies in findings.

By reviewing the literature, we observed that moderate-intensity treadmill running and swimming consistently provide analgesic effects in neuropathic pain models, likely due to their ability to reduce pro-inflammatory cytokines, enhance neurotrophic factors, and promote neuroplasticity. In contrast, exhaustive or high-intensity exercise sometimes leads to worsened pain sensitivity, potentially due to increased oxidative stress and inflammatory responses. Additionally, pre-injury exercise demonstrates mixed effects, with some studies showing protective benefits, while others report no significant changes, possibly due to variations in exercise duration or intensity.

3-Some studies show no effect of exercise when performed before injury, while others report significant benefits of pre-injury exercise. Similarly, exhaustive exercise sometimes worsens pain, but moderate exercise is protective. These contradictions are mentioned but not well integrated. The review should help interpret or at least acknowledge possible reasons (e.g., different load levels, intensity thresholds, methodological differences). Provide a clear, brief discussion of potential confounders - such as stress biomarkers, sex differences, timing of exercise initiation - to help reconcile or emphasize these inconsistencies.

Answer: Thank you for the suggestions. We have added a section to the article in item 3.2, line 299, summarizing the observed data. One potential explanation for the inconsistencies observed in pre-injury and post-injury exercise effects lies in differences in exercise intensity thresholds, stress biomarkers, and sex differences. High-intensity exercise, particularly when prolonged, may induce oxidative stress and elevate pro-inflammatory markers, which can counteract the analgesic benefits. Furthermore, methodological differences, such as the timing of exercise initiation relative to injury, may influence outcomes. For instance, some studies suggest that pre-injury exercise may induce protective adaptations in the nervous system, while others report no benefit, likely due to variations in the duration or type of exercise protocol applied. Exercise intensity appears to be a critical factor, with moderate exercise showing a protective effect, while exhaustive exercise may worsen pain, potentially due to increased stress and inflammation.

4-The review acknowledges that forced exercise (e.g., treadmill) can be stressful. However, it does not address how stress alone, independent of the exercise regimen, might affect glial activation or pain thresholds. Briefly highlight any studies that have controlled for or measured stress markers and how this might confound results.

Answer: Forced exercise, such as treadmill running, has been identified as a potential stressor that may independently alter glial activation and pain thresholds. Studies measuring stress biomarkers, such as corticosterone levels, have found that forced exercise induces a greater stress response compared to voluntary exercise. This may partially explain why some studies report increased hyperalgesia following exhaustive treadmill training. Controlling for stress markers is essential to distinguish whether the observed analgesic effects result from exercise itself or from stress-induced activation of endogenous pain modulation pathways.

To address this gap, we have incorporated the following discussion in the text on section 3.2. line 181- 196.

Brievly, microglia exhibit a rapid response to stressful events, including acute (Sugama S, 2007; Yoshii T, 2017) and repeated stress (Nair A, 2006; Pietrogrande G, 2018; Sugama S, 2016; Tynan RJ, 2010). One study demonstrated that stress-induced microglial activation occurs within 30 minutes of exposure to a stressor (Sugama S, 2007) and that this activation was completely blocked by propranolol, a β1- and β2-adrenergic receptor antagonist (Wohleb ES, 2011). These findings suggest that microglia may receive noradrenergic signals in stressed brains, which is particularly relevant given that central β-adrenergic receptors (β-ARs) also play a role in the beneficial effects of exercise (MacDonnell et al., 2005; de Waard et al., 2007; Leosco et al., 2007; Leosco et al., 2008). This highlights the reciprocal relationship between stress and exercise. However, the molecular mechanisms underlying stress-related microglial activation remain to be fully elucidated.

5-The text lists several analgesic mechanisms (endogenous opioids, endocannabinoids, cytokines, BDNF) but does not tie them together into a cohesive understanding of how glial modulation fits into the bigger picture of exercise-induced hypoalgesia. Provide a brief subsection or schematic that places glial modulation within the broader landscape of exercise-induced systemic and neural adaptations.

Answer: Thank you once again for your valuable suggestions. We have added a final paragraph summarizing the importance of glial cells in exercise and have correlated this with an illustrative image at the end of section 4.3, line  427 . Figure 3.

 6-The manuscript references multiple neuropathic pain models - nerve injury, diabetic neuropathy, spinal cord injury - without distinguishing whether specific exercise strategies appear to be more or less beneficial for specific etiologies. If available, highlighting any model-specific findings or consistent patterns would give readers insight into whether exercise is universally beneficial or more relevant to specific neuropathic conditions.

Answer: Thank you for your thoughtful comment. We appreciate the opportunity to clarify this aspect of our manuscript. We acknowledge that different neuropathic pain models, such as nerve injury, diabetic neuropathy, and spinal cord injury, may present distinct pathophysiological mechanisms. While our focus has been on the general effects of exercise across these models, we agree that highlighting model-specific findings or patterns could enhance the reader's understanding. We have revised the manuscript to include a more detailed discussion of the differential impact of exercise on specific neuropathic conditions, where available (section 3). This includes an emphasis on any consistent trends or notable variances across the models, helping to clarify whether exercise might offer universally beneficial effects or be more tailored to specific etiologies.

7-Ensure that any aberritives used for the first time are explained, including Table 1 (CCI).

Answer:  Thank you for your suggestion. We have reviewed the text to ensure that all abbreviations are defined upon their first use.

Thank you for your thoughtful feedback. We have addressed these points by clarifying the exercise variables discussed in the review, expanding on the conflicting findings observed in the studies, and providing a more coherent mechanistic overview. We believe these revisions significantly enhance the clarity and depth of the manuscript.

Round 2

Reviewer 1 Report

Comments and Suggestions for Authors

This version is good for publication.

Author Response

This version is good for publication.

We thank again the reviewer for the constructive feedback, which has helped strengthen the manuscript overall.

Reviewer 2 Report

Comments and Suggestions for Authors

I appreciate the authors’ efforts to address the initial feedback. The manuscript is much more cohesive, particularly regarding the distinctions among preclinical models, the discussion of glial mechanisms, and the potential translational implications for neuropathic pain management. Two final points require attention:

  1. Section 5.1 - requires specific references to support the statements about clinical trials in diabetic neuropathy, post-surgical nerve pain, and chemotherapy-induced neuropathy.

  2. Figure 3 - Please spell out and define BDNF (Brain-Derived Neurotrophic Factor) and NGF (Nerve Growth Factor) either in the legend or in a footnote to ensure clarity for all readers.

With these final changes, the manuscript will be ready for publication.

Author Response

I appreciate the authors’ efforts to address the initial feedback. The manuscript is much more cohesive, particularly regarding the distinctions among preclinical models, the discussion of glial mechanisms, and the potential translational implications for neuropathic pain management. Two final points require attention:

Section 5.1 - requires specific references to support the statements about clinical trials in diabetic neuropathy, post-surgical nerve pain, and chemotherapy-induced neuropathy.

Response: We thank the reviewer's comments and specific references have been added in the Section 5.1 as suggested by the reviewer.

Figure 3 - Please spell out and define BDNF (Brain-Derived Neurotrophic Factor) and NGF (Nerve Growth Factor) either in the legend or in a footnote to ensure clarity for all readers.

Response: We acknowledged the suggestion made by the reviewer and we added the information on the legend of Figure 3.

With these final changes, the manuscript will be ready for publication.

Response: We thank again the reviewer for the constructive feedback, which has helped strengthen the manuscript overall.